

# Digitizing scanning lidar measurement campaign planning

Nikola Vasiljević[1], Andrea Vignaroli[1], Andreas Bechmann[1], and Rozenn Wagner[1]

[1]Technical University of Denmark - DTU Wind Energy, Frederiksborgvej 399, Building 118-VEA, 4000 Roskilde, Denmark

**Correspondence:** Nikola Vasiljević (niva@dtu.dk)

**Abstract.** Multiple wind measurements is a way to reduce the uncertainty of wind farm energy yield assessments by reducing the extrapolation distance between measurements and wind turbines locations. A WindScanner system consisting of two synchronized scanning lidar potentially represents a cost-effective solution for multi-point measurements, especially in complex terrain. However, the system limitations and limitations imposed by the wind farm site are detrimental to the installation of scanning lidars and the number and location of the measurement positions. To simplify the process of finding suitable measurement positions and associated installation locations for the WindScanner system we have devised a campaign planning workflow. The workflow consists of four phases. In the first phase, based on a preliminary wind farm layout, we generate optimum measurement positions using a greedy algorithm and a measurement 'representative radius'. In the second phase, we create several Geographical Information System (GIS) layers of information such as exclusion zones, line-of-sight (LOS) blockage, and lidar range maps. These GIS layers are then used in the third phase to find optimum positions of the WindScanners with respect to the measurement positions considering the WindScanner measurement uncertainty. In the fourth phase, we optimize and generate trajectory through the measurement positions by applying the traveling salesman problem (TSP) on these positions. The above-described workflow has been digitized into the so-called Campaign Planning Tool (CPT) currently provided as a Python library which allows users an effective way to plan measurement campaigns with WindScanner systems. In this study, the CPT has been tested on three different sites characterized by different terrain complexity and wind farm dimensions and layouts. The CPT has shown instantly whether the whole site can be covered by one system or not.

## 1 Introduction

The development of a wind farm project begins with an assessment of the wind resources and the energy yield for the planned wind farm. Best practices recommend estimating wind resources based on local wind measurements (MEASNET, 2016). Measurement campaigns designed for wind resource assessment have historically relied on anemometers and wind vanes mounted on tall meteorological masts with the goal to measure a wind climate similar to the wind climate the wind turbines will experience during their lifetime. The local measurements produce the observed wind climate of the site. To account for the seasonal and inter-annual variations of the wind the observed wind climate is long-term corrected using long-term reference





data from a nearby meteorological station or a mesoscale model (Carta et al., 2013). The long-term corrected wind climate is then extrapolated vertically and horizontally, typically using a flow model such as WAsP (Mortensen et al., 2014) to estimate the wind resource at hub height for every wind turbine location.

The single mast approach is affordable but can cause large uncertainties. Specifically, in complex terrain (mountainous and forested areas), the spatial extrapolation becomes challenging as the topography can significantly influence the flow. The ideal scenario would be to measure the local wind climate at every planned wind turbine position. However, erecting as many masts as wind turbines would be extremely costly and in some areas impossible.

    Some large wind farm projects in complex terrain have been developed using multiple masts. Combining one fixed mast and one or several roaming profiling lidars moved to different positions during the campaign is another option. The advantage
of roaming vertical profiling lidars lies in their ability to provide affordable high altitude measurements, easiness of deployment and absence of building-permits in comparison to the masts, while data availability and inaccuracy in complex terrain (Bingöl et al., 2009) are some of their disadvantages. However, any roaming setup brings a trade-off between the number of measurement positions and the measurement duration at each location since short measurements (e.g. of 3 months) can lead to erroneous wind climate (Langreder and Mercan, 2016).

A potential solution for multi-point measurements for wind resource assessment lays in the application of scanning lidars (Krishnamurthy et al., 2013). With a measurement range of several kilometers and a beam that can be oriented freely in any direction (Vasiljevic et al., 2016), many measurement positions can be reached without moving the hardware. Especially dual-Doppler setups (i.e., two scanning lidars) can provide accurate retrieval of horizontal wind speed and wind direction (i.e., two dimensional (2D) wind vector) at many possible positions (Vasiljević et al., 2017). While scanning lidars provide a broad range
of benefits, there are also clear challenges when designing multi-lidar measurement campaigns.

    Constraints which arise from scanning lidars, atmosphere and site characteristics dictate the design process mentioned above. Indeed, the beam of scanning lidars can be steered freely, but on the other hand, it can be blocked in some directions by the terrain, vegetation or other obstacles (e.g., power lines). This impacts the positioning of scanning lidars can be placed such that there is a clear line-of-sight (LOS), i.e. unblocked passage of the beams towards measurement points. On the other hand, the
lidar characteristics (e.g., laser wavelength and output power) in combination with the atmosphere characteristics (e.g., aerosol extinction, backscatter coefficient, and atmospheric attenuation) impact the maximum expected range of the lidar. Furthermore, retrieving the 2D wind vector requires a limited beam elevation angle (e.g., smaller than 5° suggested by Vasiljević et al. (2017)) to avoid contamination of horizontal wind components with the vertical component, finally the intersecting angle of the beams at the measurement points should be large enough (e.g., bigger than 30° suggested by Vasiljevic and Courtney (2017)) to
minimise the lidar measurement uncertainty (Davies-Jones, 1979; Stawiarski et al., 2013). Overall, a campaign planner has to handle several constraints at the same time to find the best measurement locations and in accordance with them generate the best possible measurement campaign layout.

    In this paper, we describe a workflow and resulting digital tool (named Campaign Planning Tool, CPT) which tackle the above-described challenges involved in the planning of scanning lidar campaigns. The workflow is based on the application
of the methodology for multi-lidar experiments on wind resource assessment campaigns (Vasiljević et al., 2017), which was





previously used in planning of more than 20 measurement campaigns (Vasiljevic, 2018) and especially those conveyed in the New European Wind Atlas (NEWA) project (Mann et al., 2017), such as Perdigao-2015 (Vasiljević et al., 2017) and Perdigao-2017 (Fernando et al., 2019). On the other hand, the CPT has been previously conceptualized during the WindScanner.eu project (see 'WindScanner locator' description on page 8 in Vasiljevic and Hasager, 2015).

The paper is organized as follows. Section 2 describes the workflow and corresponding elements of CPT. In Section 3 we present results of applying CPT for planning campaigns at three sites. We discuss the results and future work in Section 4, while we provide our concluding remarks in Section 5.

## 2    Campaign planning workflow

### 2.1    Overview

As a starting point, the location and the layout of the wind farm are assumed to be known. The campaign planning workflow consists of four phases which are graphically represented in Figure 1:

1. The measurement positions are determined based on the wind farm layout.

2. The lidar and site (geographical) constraints are collected and combined.

3. The positions of the scanning lidars are determined.

4. The trajectory of the laser beams through all the measurement points is generated.

Each phase consists of several interconnected modules (represented as icons in Figure 1). The modules entail algorithms that have been developed in Python. Data used as inputs from modules are obtained though connections to public databases. A detailed description of the different phases and associated modules are given in the following paragraphs.

### 2.2    Phase 1 - Measurement positions

We assume that the wind farm site has been selected and that a preliminary resource assessment and wind farm layout have been made prior to the campaign planning. The wind farm layout is a required input for the campaign planning. It is used to determine the measurement positions. For small wind farms (either a limited number of turbines and-or a limited spatial extent) we can coincide the measurement positions with the wind turbine positions. For larger wind farms, the number of measurement points needs to be reduced.

The 'Measurement optimization' module optimizes the number of measurement positions and their location. The approach we have used to group the wind turbine locations, which are close to each other in clusters, and to assign a single measurement location per cluster. MEASNET (2016) suggests that measurements from a single location represent the wind climate over a certain area described by 'representativeness radius' ($R_r$). $R_r$ has different values for different terrain types. For example, in complex terrain, the radius should be smaller than 2 km. By solving a disc covering problem (e.g., Biniaz et al., 2017), in which

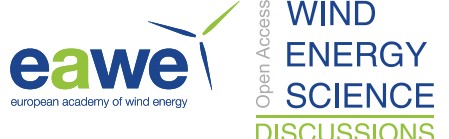

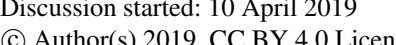

**Figure 1.** Campaign planning workflow (figure design using freepik.com icon database)



we aim to find a minimum number of discs with a radius equal to $R_r$ that cover all locations of wind turbines, we cluster the wind turbines and optimize the measurement locations. The 'Measurement optimization' module includes the greedy method implementation of the disc covering problem and outputs optimized measurement locations, which are geolocated in Universal Transverse Mercator (UTM) coordinate system.

The 'Map setup' module calculates the extent of the map for the selected site and generates a mesh over the map area. The center of the map is defined as the barycenter of the measurement locations. To calculate the relative map extent along the x- and y-axis (i.e., Easting and Northing) the sum of the distance of the barycenter to the furthest measurement point and maximum expected range of the lidar (defined in the "Lidar Range" module) is multiplied by two. This extent is added or subtracted from the x- and y- coordinates of the barycenter, yielding four corners of the map that describes a rectangle that

encompasses the wind farm site.

The second input to the 'Map setup' module is the mesh resolution, which is used together with the four calculated corners to generate a mesh over the map area. Usually, the mesh resolution is set to 100 m to match the resolution of public databases used in Phase 2. Afterward, another copy of the map corners and the mesh is made by re-projecting the UTM values to the latitude and longitude.

The outputs of Phase 1 of the campaign planning workflow are two sets of four corners describing the map area, the mesh containing equally spaced points covering the map area and the positions of the measurement points in the UTM coordinate system and latitude-longitude coordinate system. The UTM coordinate system is used in most modules since it is more intuitive to operate with a Cartesian coordinate system, whereas the latitude-longitude coordinate system is primarily used to fetch data from public databases containing geographical data.

**2.3   Phase 2 - Geographical and lidar related constraints**

Each mesh point is considered as a potential location to place one of the two lidars. The purpose of Phase 2 is to create a map indicating a number of measurement positions that can be reached by the lidars, for each mesh point, considering 5 types of constraints: zones where a lidar cannot be installed (e.g., lakes); keeping the lidar elevation angle below a certain threshold to avoid measurement contamination with the vertical component of the wind; the maximum lidar range; topographical features

that can block the beam; practical matters such as access roads.

The generated map extent, mesh, and measurement positions are used to create five Geographical Information System (GIS) layers, which aid the placement of the dual-Doppler setup (i.e., two scanning lidars). Three transient GIS layers need to be created first.

**2.3.1   Transient GIS layers**

The "Orography" module establishes a connection to the NASA server hosting digital elevation model (DEM) data from the Shuttle Radar Topography Mission (SRTM, Farr et al. (2007)). The DEM data have a horizontal resolution of 100 m. The map corners and mesh given as a set of latitudes and longitudes are used to acquire orography information. The Orography module fetches the elevation for each mesh point and creates an orography GIS layer.




The "Canopy" module acquires the canopy properties for the given site through the land cover information for the area. The land cover information is acquired either from the CORINE Land Cover dataset (for locations in Europe) or from the Global Land Cover 2000 dataset (for sites outside Europe). Both datasets are publicly available for download. The land cover data are geolocated in the UTM coordinate system; thus this module uses map corners in the UTM projection to extract a portion of the

land cover map. Since this transient GIS layer only contains the information on the type of the land cover, the Canopy module assigns heights for each land cover type based on a lookup table, which produces one more transient GIS layer (canopy height GIS layer). The look-up table is made manually. Currently, we set the look-up table such that it assigns 20 m height for areas covered by trees.

The "Topography" module creates one more transient GIS layer (topography GIS layer) by merely combining the orography

and the canopy height GIS layers.

### 2.3.2 Main GIS layers

The "Exclusion zones" module, using the land cover GIS layer, creates the first main GIS layer that indicates areas of the map where lidars cannot be installed (e.g., over the water surface, on the top of the forest). This GIS layer is saved as a GeoTIFF image. We use the GeoTIFF format since it supported by many GIS based software solutions, such as Google Earth or QGIS.

The "LOS blockage" module creates the second main GIS layer. This layer is generated by performing a viewshed analysis for measurement positions based on the topography GIS layer. Basically, the LOS blockage module assigns which measurement positions are visible from each mesh point.

The "Elevation angle" module considers each mesh point as a lidar installation location and calculates the required elevation angles to steer the laser beam towards each measurement locations, based on the transient topography GIS layer. The current

tool is mainly designed to plan measurements with two synchronized scanning lidars (dual-Doppler WindScanner system). The main goal is to retrieve the horizontal wind speed. A low elevation beam angle is required to avoid contamination of the LOS speed measurement with the vertical component of the wind vector. The module assigns which measurement positions are 'reachable' with an elevation angle below a given threshold value (e.g., 5° suggested by Vasiljević et al. (2017)). This process creates the third GIS layer.

For each mesh point, the "Lidar range" module assigns which measurement points are within reach of the lidar taking into account the expected range of the lidar for the given site. The Lidar range module makes use of the orography transition GIS layer and positions of the measurement points (as well as their height) when performing the underlying calculations.

Finally, the fifth main GIS layer is a geolocated satellite image matching the desired area of the map (i.e., the wind farm site). The 'Satellite image' module, based on the map corners given in terms of latitudes and longitudes, compiles a list of requests

for the Google map server. These requests are pushed through Google's Maps Static Application Programming Interface (API) and result in the acquisition of a set of tiles (satellite images) that cover the map area. Once all the tiles are fetched, the module assembles them in a single aerial photo of the site. Afterward, the module geolocates the aerial photo in the UTM coordinates as a GeoTIFF file. The satellite image is used in Phase 3 to identify access roads and possible power source.



## 2.4 Phase 3 - Placement of the lidars

Phase 3 provides adequate locations for two scanning lidars working as a dual-Doppler system. Basically, the combination of the previous GIS layers highlights the 'best' locations for the placement of individual lidars considering all the above-described constraints. However, designing the campaign for a dual-Doppler system, where beams from two lidars need to

synchronously cross at every measurement positions, adds one more constraint: the limitation on the beams intersection angle. The measurement uncertainty of a dual-Doppler system increases when the intersecting angle between the laser beam gets small (see Vasiljevic and Courtney (2017)). Therefore the position of the second lidar is very much determined by the position of the first lidar.

The "Combine Layer" module provides a map indicating all possible positions for the first lidar accounting for the geograph-

ical and lidar constraints described in Phase 2. The satellite image layer is used as background and we overlay the combination of the other four GIS layers as one single layer. The overlaid layer, which will be referred to as the combined layer (CB), acts as a constrainer for the lidar placement. Specifically, the CB layer is made of exclusion zones (EZ), LOS blockage (LB), elevation angle (EA) and lidar range (LR) layers.

To create the CB layer, we intersect the information from the EZ, LB, EA and LR layers at each mesh point. For each mesh

point, layer LB, EA, and LR contain either an empty set or a set containing IDs of reachable measurement points. For each mesh point, the EZ layer contains a single value indicating whether it is possible or not to install a lidar (value equal to 1 and 0 respectively). To create the CB layer, we use the following formula:

$$CB(i) = \begin{cases} \{\}, & \text{if } EZ(i) = 0 \\ LB(i) \cap EA(i) \cap LR(i), & \text{otherwise} \end{cases}$$

where $i$ represents an index of the mesh point. If $EZ(i)$ is equal to zero, for mesh point $i$, the produced set for CB layer is be

empty; otherwise, it contains the intersection between three sets (i.e., three GIS layers).

The user needs to choose one of the possible locations for the first lidar (from the CB layer). The 'First lidar placement' module finds the mesh point ID, which corresponds to the first lidar position and fetches the IDs of the visible measurement points from the CB layer. The measurement points IDs and the first lidar positions are then supplied to the 'Intersecting angle' module.

Then, the "Intersecting angle" module considers that each mesh point is a potential location for the second lidar placement and performs the following tasks:

1. Calculates the intersecting angles that the two laser beams will have at the measurement positions indicated by the IDs set generated by the First Lidar Placement module;

2. Creates a set containing IDs of the measurement points for which the intersecting angle is higher than a specific value
(e.g., 30° suggested by Vasiljevic and Courtney (2017));

3. Intersects the set above with the corresponding set from the CB layer;





4. Saves the intersected set for each mesh point creating a new GIS layer to which we will refer to as the intersecting angle (IA) GIS layer.

In this way, the IA layer highlights the best locations for the placement of the second lidar containing, besides the preliminary set of constraints, the constraint on the measurement uncertainty (indirectly via the intersecting angle threshold). This new GIS

layer aids the campaign planner in selecting the location for the second lidar. Once the position of the second lidar is chosen the process of generating a potential lidar installation layout is completed. The output of the module 'Second lidar placement' are the positions of the two lidars and IDs of measurement points, which are 'measurable', i.e., visible by the two lidars considering all the constraints.

It is important to highlight that the CB layer can provide several possible positions for the first lidar. In Phase 3, the workflow

user needs to consider every possible position for the first lidar and can, for each of them, run the "First lidar placement" and the "Intersecting Angle" modules to identify the possible positions for the second lidar. For most sites, several measurement campaign layouts can be generated. It is advisable to generate several potential layouts, since only during a field visit will it be possible to determine the most likely design for the measurement campaign.

## 2.5 Phase 4 - Trajectory generation

The fourth phase consists of the optimization of the path through the measurement points and the generation of the corresponding trajectory.

In the previous phases, we derived the measurement locations and dual-Doppler campaign layout(s). A next task is to optimize the path through those positions such that the motion of the scanner heads required to steer the beams takes the least amount of time (i.e., increasing sampling rate). One way to achieve this is to adapt the solution for the traveling salesman

problem (TSP). In the regular TSP, the goal is to find the shortest path through a set of $n$ cities that a traveling salesman needs to visit. In our case, we have two 'salesmen' (i.e., lidars), and two set of cities because the two lidars do not have identical locations. To level our problem of the trajectory optimization to the regular TSP problem we need to convert two 'salesmen' and two sets of 'cities' to a single salesman and single set of cities. To achieve this, we do the following steps:

1. We create two arrays, one containing measurement points visible by both WindScanners, and second one corresponding

to the trajectory which will be empty.

2. From the measurement point array, we randomly select one of the points, add it to the trajectory array and then remove the same point from the measurement point array. At the end of this step, the trajectory array contains one measurement point.

3. Next, we calculate the required relative angular moves that two lidars would need to perform from the current element

of the trajectory array to reach any remaining measurement point in the measurement point array. This forms two arrays containing angular moves corresponding to the two lidars.



4. In the next step, we form a new array containing the maximum value for the pairs of the angular moves from the two above-described arrays. This step converts our problem of optimizing the path through the measurement points to the general TSP problem, since now we have single set of cities and single traveling salesman.

5. Next, we find the element which has minimum value in the maximum angular move array based. The corresponding measurement point for this element represents the next trajectory point, which is then added to the trajectory array and removed from the measurement point array.

6. We repeat steps 3 to 5 until the measurement point list is empty.

The above-described steps are encapsulated as an algorithm, which is a fundamental block of the 'Trajectory optimization' module. The output of the module is an efficient order to probe the measurement points in space with the two laser beams steered by the lidar scanner heads.

To get the lidars to actually follow the trajectory, we need to describe the motion of the scanners as a function of time. In other words, we need to 'attach' the time component of the trajectory to the spatial description we yielded in the previous steps. We aim at minimizing the time required to move from one measurement point to another. Since we derived the order of measurement points to do this, we need to know the kinematic limits of the scanner heads, specifically maximum speed and maximum acceleration. These two parameters along with the required angular move that the scanner head of each lidar needs to do to steer the laser beams from one to another measurement point are used to solve the kinematics elevator problem (e.g., Al-Sharif, 2014). The solution for this problem yields the minimum required time to move a scanner head from one to another position. Since we have two lidars that move to a measurement point, we will generally have two different moving times. To keep the lidar measurements in sync in both time and space, we take the maximum of the two derived values. When calculating the trajectory, we assume that the lidars will stop at each measurement point and sample wind speed before they continue to the next measurement points. Therefore, we expect that lidars will perform so-called step-stare trajectories. There are several reasons for selecting step-stare trajectories instead of continuously scanning the flow through the trajectory described by the measurement points. The most important reason is that the current commercial scanning lidars allow only step-stare implementation of complex trajectories. Furthermore, the application of the continuously scanning through complex trajectories is not trivial and, it requires more complex kinematic models than the one described by the elevator problem.

## 3 Campaign planning workflow in action

### 3.1 Overview

In this section, the campaign planning workflow is demonstrated through the application of CPT on three different wind farm sites. The only information needed for each site is the wind turbine positions and their hub height. This input could be generated arbitrarily, but to make the examples realistic actual operating wind farms have been chosen. The three selected sites are all located in complex terrain, where large spatial variations in wind speed are expected, and the sites are thus relevant for scanning




lidar campaigns. The spatial extent of the sites varies greatly: with a single centrally placed met mast the maximum distance to a turbine would be 1 km, 4 km and 5 km for the three sites. Since only the wind farm layout and the turbine hub height are needed for the demonstration, any other details that could identify the wind farms are omitted in the descriptions below. The wind farms are just named by their country of origin: Scotland, Turkey, and Italy.

For all three sites, we aim to design the campaign for the long-range WindScanner system configured in a dual-Doppler mode (i.e., the system will have two scanning lidars). The system is described in details in Vasiljevic et al. (2016). To demonstrate the workflow, the most essential bits of information is the maximum range of the lidars, which is 6 km, and maximum velocity and acceleration of the scanner heads, which are $50°/s$ and $100°/s^2$ respectively.

## 3.2   Site 1 - Scotland

The Scottish site consists of 22 wind turbines with 47-m hub-heights and has a quite compact layout (Figure 2). The distance between adjacent turbines is about 300 m (5 rotor diameters). The wind farm is placed on a 300-m tall hill surrounded by rolling hills of farmland with windbreaks and patches of forest. The hill is quite steep with maximum slopes of 20% from the main south-western wind direction. The site is located 17 km from the coast and can, therefore, be considered an inland site.

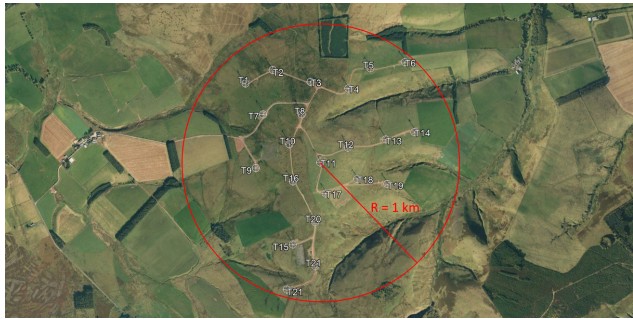

**Figure 2.** Google Earth image of the Scottish site. A 1 km radius circle illustrates the extent of the wind farm

Due to the compact design of the wind farm, we decided to skip the measurement position optimization and try to generate
a measurement campaign in which we intend to measure at every wind turbine position. Figure 3 shows the map extent and locations of the wind turbines, now measurement positions, generated by the 'Map Setup' module. Considering that the site is relatively close to the coast, surrounded by agricultural land, and the altitude is about 300 m above sea level (asl.), thus relatively low, the site should experience a good concentration of aerosols. Nevertheless, we cannot expect that the WindScanners will have 6 km range all the time and assume that on average the WindScanners would have a range of at least 3 km at the selected
site (i.e., half of the maximum claimed range). This estimation is based on our experience in doing measurement campaigns at various locations and in different atmospheric conditions.

Using this range together with the map extent, we generated the CB layer (see top image in Figure 4). The dark red color areas show positions from where an individual scanning lidar can reach out to all measurement positions. Those areas are relatively





large because the wind farm layout is compact. For the purpose of this example, we chose to place the first WindScanner at the South of the wind farm (coordinates of 100 m, -1600m and 350 m in Easting, Northing and altitude asl. respectively).

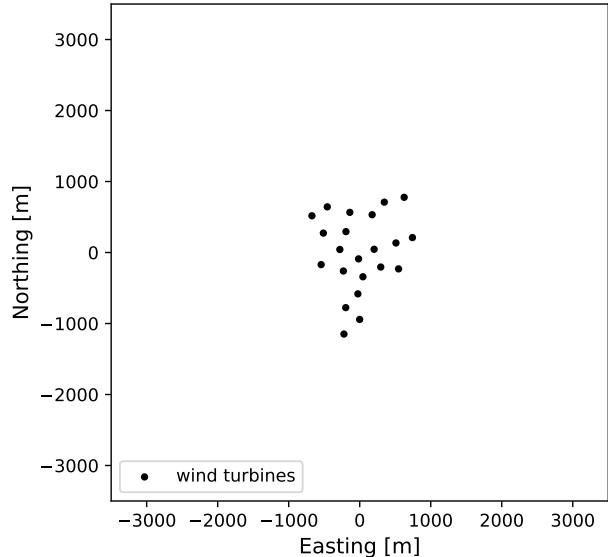

**Figure 3.** Measurement locations for Scottish site: black dots - wind turbine positions that also correspond to measurement positions.

As explained in Section 2 (Phase 3), the first lidar placement is detrimental for the second lidar placement because of the intersecting angle between the respective lidars' beams. There is only one area of the map where the placement of the

second lidar assures that all measurement points are within reach and measurable with fair accuracy (bottom image in Figure 4). By selecting the position of the second lidar (coordinates of 1600 m, 600m and 318 m in Easting, Northing and altitude asl. respectively), we complete the generation of one measurement campaign layout. In practice, we would generate several layouts (for different positions of WindScanner 1 and WindScanner 2), and assess their feasibility by inspecting aerial images, e.g. looking for access roads and nearby power lines or houses. However, for the sake of demonstrating the workflow, we have

generated only a single layout.

Since we have both the measurement and lidar positions, we have all the elements to optimize and generate the trajectory. Figure 5 shows the optimum trajectory through the measurement points, resulting from the modified TSP (see Section 2 - Phase 4). The second column of Table 1 and Table 2 show the trajectory order and angular positions respectively.

Considering the kinematic limits of the scanner head and that we are performing step-stare scans, we can apply the elevator

kinematic problem on the trajectory points (Table 2). This step yields the required time to move the scanner heads from one point of the trajectory to another. The input to the elevator kinematic problem is the foreseen angular displacement, maximum velocity, and acceleration of the scanner head. In our case, we have two angular movements for each measurement point (see Table 2 and 3) since the scanner heads will move in both azimuth and elevation axis since the measurement points do not lay on the same altitude. However, the displacement in the azimuth angle $\theta$ is much larger than the one in the elevation angle $\varphi$, and





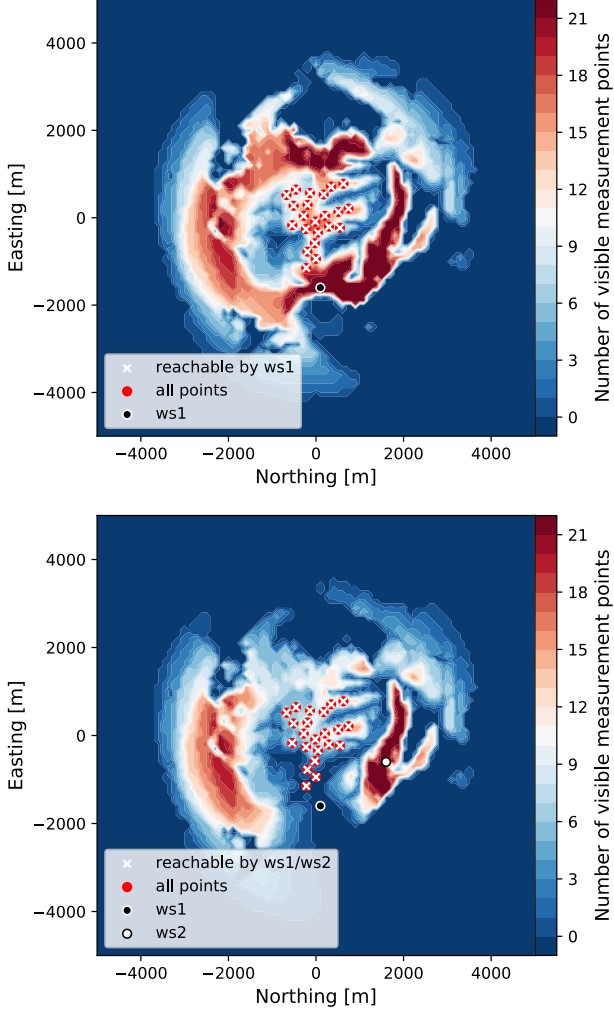

**Figure 4.** Placing lidars at the Scottish site: top image - locating first lidat at the CB layer, bottom image - locating second lidar at IA layer

it will dictate the minimum time for the scanner head motion (see Table 2). Therefore, we use the displacement in the azimuth angle as an input for the kinematic model (see the second and third column in Table 3). The kinematic model calculates the minimum time to perform the move (see the third and fifth column in Table 3). As we can see in Table 3 the minimum time for each WindScanner will be different. To keep the WindScanners synchronized, we use the maximum of the two calculated

5     values for each trajectory point (last column in Table 3). At each point in the trajectory, the WindScanners will accumulate spectra over a period of 1 s. Therefore, one complete scan of all measurement points will take about 36 s, of which 22 s is for measurements while the remaining amount is for the motion between the measurement points, which results in about 16 samples of each measurement point per 10-min period. Typically we aim at having at least ten samples per 10-min period.





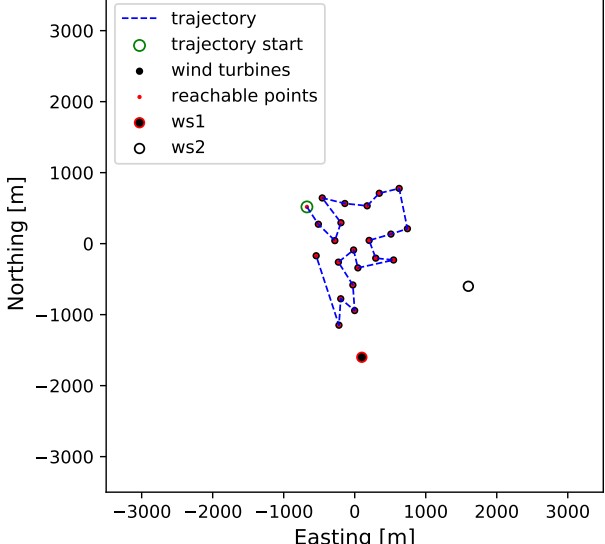

**Figure 5.** Final campaign design at Scottish site

### 3.3   Site 2 - Italy

The Italian wind farm consists of 36 wind turbines with a 78-m hub-height. The turbines are distributed over a large area (see Figure 6) but somehow clustered in small groups (Figure 11) often with inter-turbine distances of less than 300 m (3 rotor diameters). With a coastline only 10 km to the West, a complex coastal-inland wind climate transition is expected to occur across the wind farm. The terrain has an average 7% slope from the coast to the wind farm. The wind farm is surrounded by farmland, though in a range of about 7 km there are several medium-size towns forming an urban area ring around the farm site.

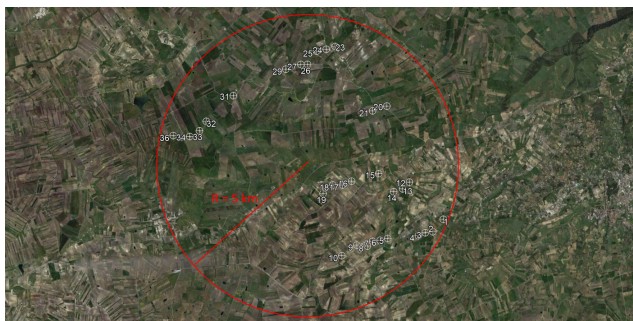

**Figure 6.** The aerial image of the Italian site (source Google Earth). A 5 km radius circle illustrates the large extent of the wind farm





**Table 1.** Measurement points at the Scottish site. All position values rounded to two decimals.

| Initial order | Trajectory order | Easting [m] | Northing [m] | Height [m] | Visible by WS1 | Visible by WS2 |
|---|---|---|---|---|---|---|
| 1 | 1 | -673.0 | 517.5 | 323.0 | True | True |
| 2 | 5 | -457.0 | 643.5 | 321.0 | True | True |
| 3 | 6 | -139.0 | 565.5 | 335.0 | True | True |
| 4 | 7 | 175.0 | 532.5 | 346.0 | True | True |
| 5 | 8 | 346.0 | 709.5 | 341.0 | True | True |
| 6 | 9 | 626.0 | 777.5 | 327.0 | True | True |
| 7 | 2 | -512.0 | 273.5 | 338.0 | True | True |
| 8 | 4 | -194.0 | 294.5 | 358.0 | True | True |
| 9 | 22 | -543.0 | -170.5 | 347.0 | True | True |
| 10 | 3 | -280.0 | 42.5 | 370.0 | True | True |
| 11 | 16 | -16.0 | -89.5 | 383.0 | True | True |
| 12 | 12 | 203.0 | 45.5 | 372.0 | True | True |
| 13 | 11 | 510.0 | 134.5 | 349.0 | True | True |
| 14 | 10 | 742.0 | 210.5 | 335.0 | True | True |
| 15 | 20 | -197.0 | -776.5 | 352.0 | True | True |
| 16 | 17 | -230.0 | -260.5 | 380.0 | True | True |
| 17 | 15 | 45.0 | -341.5 | 383.0 | True | True |
| 18 | 13 | 296.0 | -204.5 | 366.0 | True | True |
| 19 | 14 | 547.0 | -230.5 | 338.0 | True | True |
| 20 | 18 | -26.0 | -582.5 | 363.0 | True | True |
| 21 | 19 | -1.0 | -942.5 | 351.0 | True | True |
| 22 | 21 | -222.0 | -1148.5 | 351.0 | True | True |

Given the specific layout of the wind farm, having more or less isolated groups of tightly packed wind turbines, we decided to apply the measurement point optimization. For this wind farm, the representativeness radius was set to 500 m, which is four times smaller than the value suggested for the complex terrain sites (MEASNET, 2016). With this conservative setting, the optimization routine found 13 discs of radius equal to 500 m which covers all 36 wind turbine locations (Figure 11). The disc centers are measurement positions. The disc centers coordinates are listed in Table 4.

From there, the workflow was applied in the same way as it was for the Scottish site. In comparison to the Scottish site, the Italian wind farm is even closer to the sea, and it is surrounded by an urban area that in our experience increases the aerosol concentration resulting in an improved lidar range. Therefore, for the Italian site, we can expect to have an average measurement range of 4 km for the WindScanners. The CB layer generated by the CPT is shown as the top image in Figure 8. For this site, there are actually no positions from which a lidar can reach all 13 measurement positions (in spite of the 4





**Table 2.** Angular positions for WindScanners for the Scottish site. All values rounded to two decimals.

| Trajectory points | $\theta_{ws1}$ [°] | $\varphi_{ws1}$ [°] | $\theta_{ws2}$ [°] | $\varphi_{ws2}$ [°] |
|---|---|---|---|---|
| 1 | 339.95 | -0.69 | 296.19 | 0.11 |
| 2 | 341.91 | -0.35 | 292.48 | 0.5 |
| 3 | 346.98 | 0.68 | 288.88 | 1.5 |
| 4 | 351.18 | 0.24 | 296.51 | 1.14 |
| 5 | 346.06 | -0.72 | 301.16 | 0.07 |
| 6 | 353.7 | -0.39 | 303.84 | 0.47 |
| 7 | 2.01 | -0.11 | 308.49 | 0.88 |
| 8 | 6.08 | -0.22 | 316.25 | 0.73 |
| 9 | 12.47 | -0.54 | 324.75 | 0.31 |
| 10 | 19.52 | -0.45 | 313.39 | 0.82 |
| 11 | 13.3 | -0.03 | 303.99 | 1.35 |
| 12 | 3.58 | 0.76 | 294.82 | 2.01 |
| 13 | 7.99 | 0.65 | 286.89 | 2.02 |
| 14 | 18.07 | -0.48 | 289.36 | 1.03 |
| 15 | 357.5 | 1.5 | 279.46 | 2.36 |
| 16 | 355.61 | 1.25 | 287.55 | 2.2 |
| 17 | 346.17 | 1.25 | 280.53 | 1.91 |
| 18 | 352.94 | 0.73 | 270.63 | 1.59 |
| 19 | 351.27 | 0.09 | 257.94 | 1.15 |
| 20 | 340.18 | 0.13 | 264.41 | 1.08 |
| 21 | 324.53 | 0.1 | 253.26 | 0.99 |
| 22 | 335.79 | -0.11 | 281.35 | 0.76 |

km assumed measurement range and the reduced number of measurement points). At best, there are only two locations from which one lidar can reach 11 out of 13 measurement points. The top image of Figure 8 shows the selected location for the first lidar installation (coordinates of -910 m, -640 m and 227 m Northing, Easting and height asl. respectively).

The IA layer (the bottom image in Figure 8) shows that the second lidar can only reach 7 measurement positions at most and this can only be achieved from a few locations. Of these locations, we selected one which assures that we cover the largest extent of the wind farm. In other words, instead of measuring at positions which correspond to closely located wind turbine clusters we probe the wind resources across nearly the entire site and thus getting better spatial information on the farm wind resources. The coordinates of a selected location for the second lidar are 1600 m, 110 m and 278 m in Northing, Easting and height asl. (the bottom image in Figure 8).



**Table 3.** Result of applying elevator kinematic problem on trajectory points for the Scottish site: step-stare order - indicate motion from one to another trajectory point ,$\Delta\theta_{ws}$ - angular displacement in azimuth angle ($\theta$), $\Delta T_{ws}$ - minimum required time to complete the angular motion. All values rounded to two decimals.

| Step-stare order | $\Delta\theta_{ws1}$ [°] | $\Delta T_{ws1}$ [ms] | $\Delta\theta_{ws2}$ [°] | $\Delta T_{ws2}$ [ms] | Max($\Delta T_{ws1}$,$\Delta T_{ws2}$) [ms] |
|---|---|---|---|---|---|
| 1->2 | 1.96 | 280 | 3.71 | 385 | 385 |
| 2->3 | 5.06 | 450 | 3.60 | 379 | 450 |
| 3->4 | 4.20 | 410 | 7.63 | 552 | 552 |
| 4->5 | 5.12 | 453 | 4.65 | 431 | 453 |
| 5->6 | 7.64 | 553 | 2.68 | 327 | 553 |
| 6->7 | 8.31 | 577 | 4.65 | 431 | 577 |
| 7->8 | 4.06 | 403 | 7.76 | 557 | 557 |
| 8->9 | 6.39 | 506 | 8.50 | 583 | 583 |
| 9->10 | 7.05 | 531 | 11.36 | 674 | 674 |
| 10->11 | 6.22 | 499 | 9.39 | 613 | 613 |
| 11->12 | 9.72 | 624 | 9.18 | 606 | 624 |
| 12->13 | 4.41 | 420 | 7.92 | 563 | 563 |
| 13->14 | 10.08 | 635 | 2.47 | 314 | 635 |
| 14->15 | 20.57 | 907 | 9.9 | 629 | 907 |
| 15->16 | 1.89 | 275 | 8.09 | 569 | 569 |
| 16->17 | 9.44 | 614 | 7.02 | 530 | 614 |
| 17->18 | 6.78 | 521 | 9.89 | 629 | 629 |
| 18->19 | 1.67 | 258 | 12.69 | 712 | 712 |
| 19->20 | 11.09 | 666 | 6.46 | 508 | 666 |
| 20->21 | 15.64 | 791 | 11.15 | 668 | 791 |
| 21->22 | 11.25 | 671 | 28.09 | 1110 | 1110 |
| 22->1 | 4.16 | 408 | 14.84 | 770 | 770 |

Considering the positions of WindScanners, reachable measurement points, and kinematic limits, we derived an optimum trajectory through the measurement points and calculated the timing for the synchronized scanner head motions (see Figure 9 and Table 4 - 6). Based on the calculated timing for the scanner heads motion and considering one second accumulation time per measurement point, one scan through all the points takes roughly 21 s of which 7 s are spent on measurements. This provides us with about 28 measurement samples at each measurement point within a 10-min period.




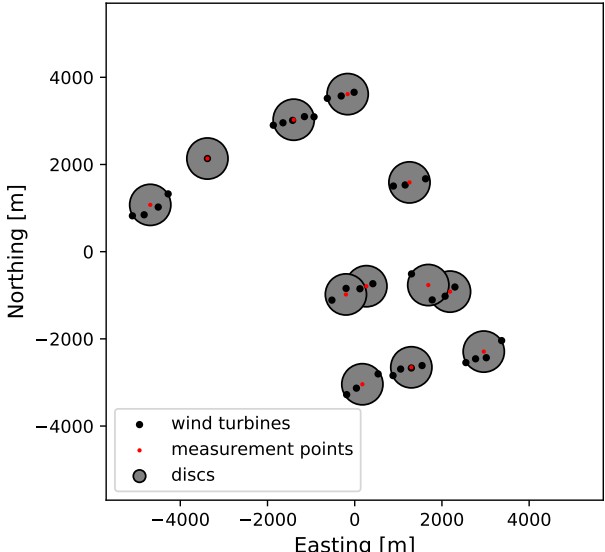

**Figure 7.** Measurement locations for Italian site: black dots - wind turbine positions, red circles - discs covering wind turbine positions, green dots - optimized measurement positions (i.e., discs' centers)

**Table 4.** Measurement points at the Italian site. All position values rounded to two decimals.

| Initial order | Trajectory order | Easting [m] | Northing [m] | Height [m] | Visible by WS1 | Visible by WS2 |
|---|---|---|---|---|---|---|
| 1 | 1 | -1019.32 | 2953.88 | 384.0 | True | True |
| 2 | | 3338.73 | -2365.62 | 336.0 | False | False |
| 3 | 5 | 1678.28 | -2726.12 | 357.0 | True | True |
| 4 | | -4308.87 | 1000.38 | 229.0 | True | False |
| 5 | | -4556.37 | 883.38 | 243.0 | True | False |
| 6 | 7 | 2564.83 | -989.62 | 407.0 | True | True |
| 7 | | 2066.98 | -839.12 | 381.0 | True | False |
| 8 | 2 | 1635.38 | 1515.38 | 344.0 | True | True |
| 9 | 4 | 647.38 | -866.12 | 352.0 | True | True |
| 10 | 6 | 555.33 | -3115.12 | 323.0 | True | True |
| 11 | | 217.83 | 3540.38 | 308.0 | False | True |
| 12 | 3 | 177.88 | -1054.62 | 328.0 | True | True |
| 13 | | -2998.02 | 2062.88 | 244.0 | True | False |

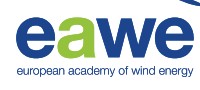



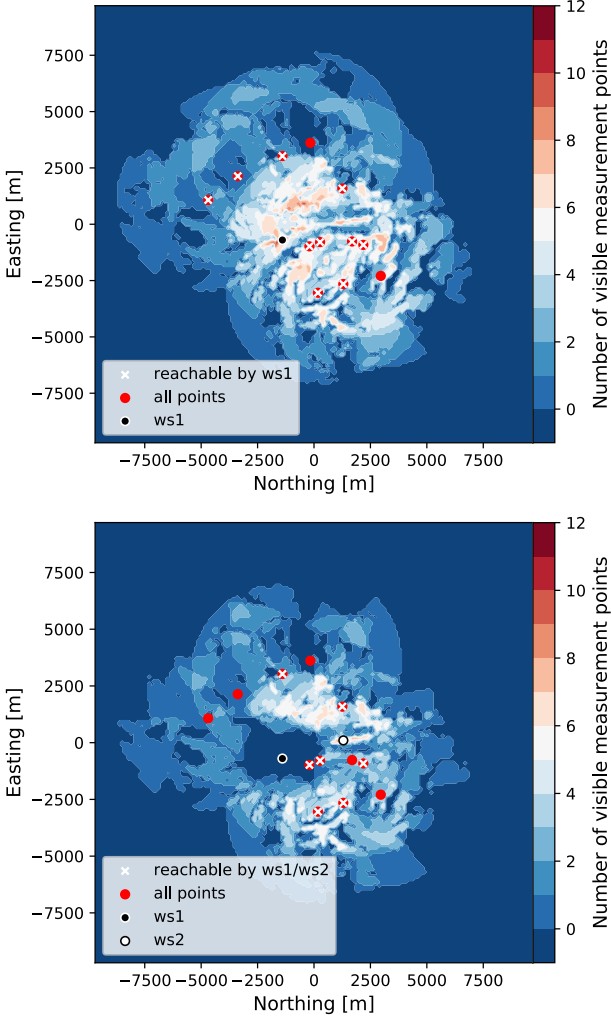

**Figure 8.** Placing lidars at Italian site: top image - locating first lidat at the CB layer, bottom image - locating second lidar at IA layer

### 3.4 Site 3 - Turkey

The Turkish wind farm consists of 22 wind turbines with a 80-m hub-height. The wind farm extends 8 km from North to South (see Figure 10) with the three most northerly turbines separated by about 2 km from the rest. The inter-turbine distance is 400-500 m (4-5 rotor diameters) for most turbines. The turbines are located along a 1600 m tall North-South ridge and the main

5   wind direction from North-East (i.e., perpendicular to the ridge line). In the main wind direction the mean terrain slopes are about 12% and with extremes reaching 50% the site should be regarded as very complex. The land cover is sparse vegetation with a patch of forest along Western facing slopes.

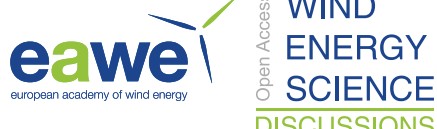



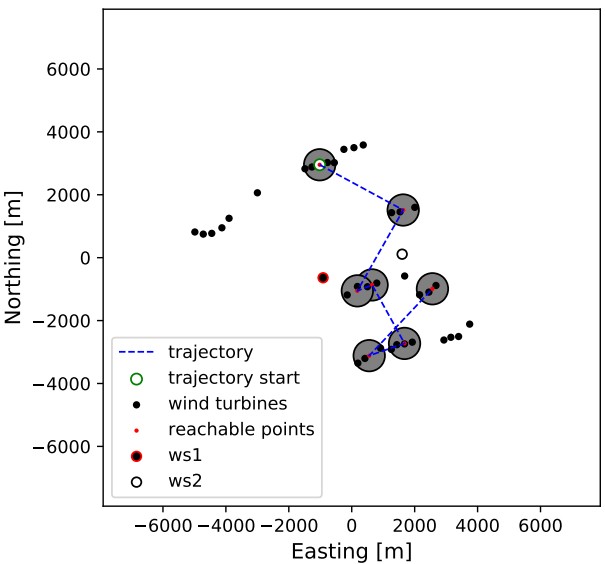

**Figure 9.** Final campaign design for Italian site

**Table 5.** Angular positions for WindScanners for the Italian site. All values rounded to two decimals.

| Trajectory points | $\theta_{ws1}$ [°] | $\varphi_{ws1}$ [°] | $\theta_{ws2}$ [°] | $\varphi_{ws2}$ [°] |
|---|---|---|---|---|
| 1 | 358.27 | 2.5 | 317.32 | 1.57 |
| 2 | 49.76 | 2.01 | 1.34 | 2.69 |
| 3 | 110.89 | 4.95 | 230.71 | 1.56 |
| 4 | 98.29 | 4.54 | 224.35 | 3.1 |
| 5 | 128.87 | 2.24 | 178.47 | 1.59 |
| 6 | 149.37 | 1.91 | 197.98 | 0.76 |
| 7 | 95.76 | 2.95 | 138.83 | 5.04 |

For this site, we assumed the average lidar measurement range to be 3 km, and we used the representativeness radius of 400 m. Our assumption on the average range in case of the Turkish site is probably closer to what a lidar would probably achieve in field operation (thus less conservative) due to operation in high altitude where we usually experience low aerosol concentration and often low clouds and fog. On the other hand, the selected representative radius is 100 m lower than in the case of the Italian site, thus about 5 times smaller than the recommended value by MEASNET. Running the workflow with using these parameters, the "Measurement optimization" module output a measurement layout with 10 measurement positions (see Figure 11 and Table 7) and Phase 2 of the workflow resulted in the CB layer shown in Figure 12, top image.



**Table 6.** Result of applying elevator kinematic problem on trajectory points for the Italian site: step-stare order - indicate motion from one to another trajectory point ,$\Delta\theta_{ws}$ - angular displacement in azimuth angle ($\theta$), $\Delta T_{ws}$ - minimum required time to complete the angular motion. All values rounded to two decimals.

| Trajectory order | $\Delta\theta_{ws1}$ [°] | $\Delta T_{ws1}$ [ms] | $\Delta\theta_{ws2}$ [°] | $\Delta T_{ws2}$ [ms] | Max($\Delta T_{ws1}$,$\Delta T_{ws2}$) [ms] |
|---|---|---|---|---|---|
| 1->2 | 51.49 | 1535 | 44.02 | 1427 | 1535 |
| 2->3 | 61.13 | 1764 | 130.63 | 3136 | 3136 |
| 3->4 | 12.6 | 710 | 6.36 | 504 | 710 |
| 4->5 | 30.58 | 1156 | 45.88 | 1455 | 1455 |
| 5->6 | 20.5 | 906 | 19.51 | 883 | 906 |
| 6->7 | 53.61 | 1614 | 59.15 | 1688 | 1688 |
| 7->1 | 97.49 | 2475 | 178.49 | 4072 | 4072 |

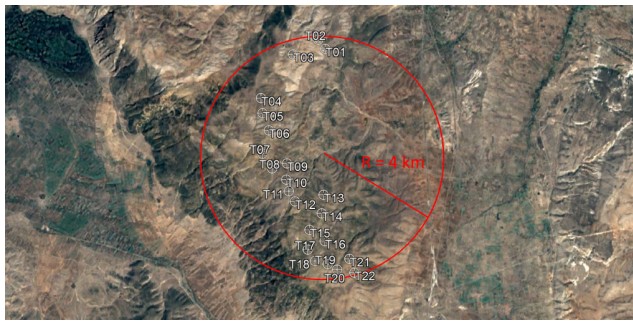

**Figure 10.** Google Earth image of the Turkish site. A 4 km radius circle illustrates the extent of the wind farm

Like for the Italian site case there are only a few locations for placing the two lidars, especially for two inter-dependent reasons one being the wind farm length (8 km) and second being the average range (3 km). Once again the best solution is to place the lidars in the middle of the wind farm. The top image of Figure 12 shows the first lidar placement, which coordinates are -1900 m, -800 m and 1497 m in Northing, Easting and height asl. respectively.

5    Knowing the first lidar position leads us to the creation of the IA GIS layers which is used for the second lidar placement. From the bottom image in Figure 12 there is only a narrow area in the middle of the wind farm where the second lidar can be placed. Also, the bottom image in Figure 12 shows the result of our choice for the second lidar placement (second lidar coordinates are -400 m, -300 m and 1569 m in Northing, Easting and height asl.).

The designed WindScanner layout can provide measurements in 6 out of 10 measurement points which cover the middle 10    part of the wind farm (Figure 13). The upper and lower quarter of the wind farm area are not reachable with the current layout. In principle, we would probably need two WindScanner systems to cover the entire wind farm (i.e., four scanning lidars). Considering the WindScanners and measurement locations together with the kinematic limits as the input for the last phase of



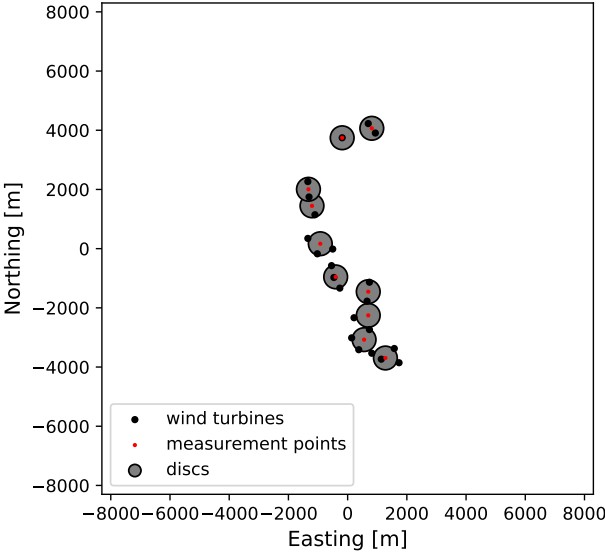

**Figure 11.** Measurement locations for the Turkish site: black dots - wind turbine positions, red circles - discs covering wind turbine positions, green dots - optimized measurement positions (i.e., discs' centers)

**Table 7.** Measurement points at the Turkish site. All position values rounded to two decimals.

| Initial order | Trajectory order | Easting [m] | Northing [m] | Height [m] | Visible by WS1 | Visible by WS2 |
|---|---|---|---|---|---|---|
| 1 | | 1276.0 | -3694.0 | 1633.0 | False | False |
| 2 | 5 | 696.0 | -2254.0 | 1676.0 | True | True |
| 3 | | 556.0 | -3074.0 | 1665.0 | False | False |
| 4 | 4 | -404.0 | -954.0 | 1613.0 | True | True |
| 5 | 3 | -924.0 | 166.0 | 1610.0 | True | True |
| 6 | | 816.0 | 4066.0 | 1551.0 | False | False |
| 7 | 6 | 696.0 | -1454.0 | 1633.0 | True | True |
| 8 | 2 | -1204.0 | 1446.0 | 1687.0 | True | True |
| 9 | 1 | -1324.0 | 2006.0 | 1734.0 | True | True |
| 10 | | -184.0 | 3746.0 | 1643.0 | False | False |

the workflow we reach the optimized trajectory which total time is 16 s of which 6 s are spent on the wind speed measurements (see Table 8 and 9). This trajectory would provide us with about 35 samples of each measurement point within a 10-min period.



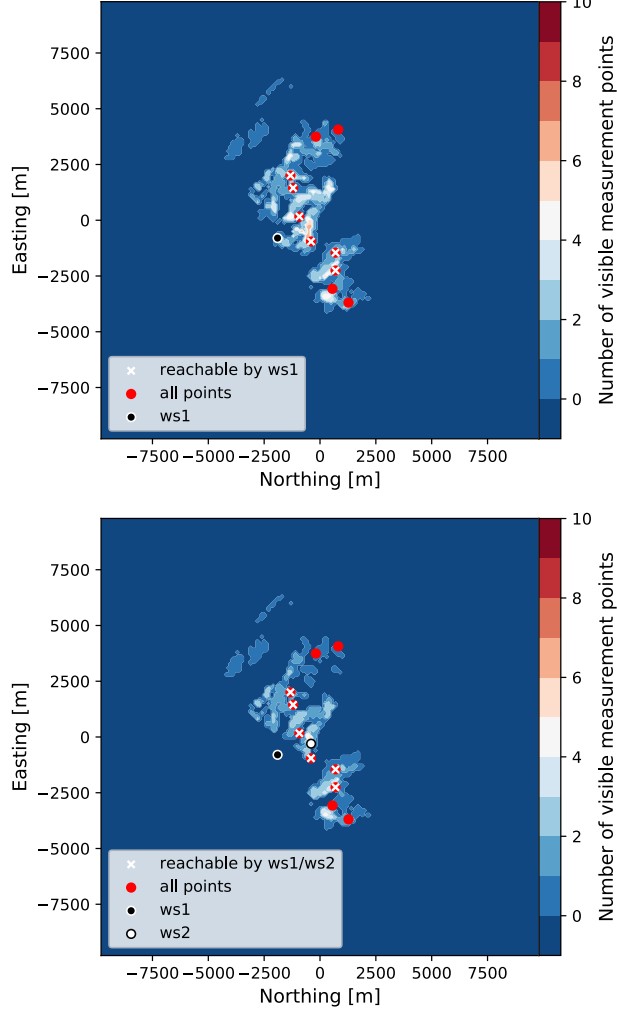

**Figure 12.** Placing lidars at the Turkish site: top image - locating first lidat at the CB layer, bottom image - locating second lidar at IA layer

## 4 Discussion

### 4.1 Discussing results

The primary purpose of the Python script up to date is to design a measurement campaign for wind resource assessment (WRA) using a long-range WindScanner system (Vasiljevic et al., 2016) configured in a dual-Doppler mode. This scope follows the RECAST project ambition which is focused on developing a new way of measuring the wind over a site for resource assessment, based on multiple measurement points using WindScanners. This has driven the choice of examples for Section 3 of this paper. However, the Campaign Planning Tool described in this paper is not limited to only planning WRA campaigns. It can be used to design any campaign using one or several scanning lidars. It can easily be applied to any type of





**Table 8.** Angular positions for WindScanners for the Turkish site. All values rounded to two decimals.

| Trajectory points | $\theta_{ws1}$ [°] | $\varphi_{ws1}$ [°] | $\theta_{ws2}$ [°] | $\varphi_{ws2}$ [°] |
|---|---|---|---|---|
| 1 | 11.6 | 4.83 | 338.16 | 3.8 |
| 2 | 17.22 | 4.74 | 335.27 | 3.51 |
| 3 | 45.3 | 4.91 | 311.65 | 3.35 |
| 4 | 95.88 | 4.6 | 180.35 | 3.85 |
| 5 | 119.25 | 3.54 | 150.71 | 2.73 |
| 6 | 104.14 | 3.01 | 136.48 | 2.3 |

**Table 9.** Result of applying elevator kinematic problem on trajectory points for the Turkish site: step-stare order - indicate motion from one to another trajectory point ,$\Delta\theta_{ws}$ - angular displacement in azimuth angle ($\theta$), $\Delta T_{ws}$ - minimum required time to complete the angular motion. All values rounded to two decimals.

| Trajectory order | $\Delta\theta_{ws1}$ [°] | $\Delta T_{ws1}$ [ms] | $\Delta\theta_{ws2}$ [°] | $\Delta T_{ws2}[ms]$ | Max($\Delta T_{ws1}$,$\Delta T_{ws2}$) [s] |
|---|---|---|---|---|---|
| 1->2 | 5.62 | 474 | 2.89 | 340 | 474 |
| 2->3 | 28.08 | 1110 | 23.63 | 972 | 1110 |
| 3->4 | 50.58 | 1522 | 131.3 | 3142 | 3142 |
| 4->5 | 23.38 | 967 | 29.64 | 1139 | 1139 |
| 5->6 | 15.11 | 777 | 14.24 | 755 | 777 |
| 6->1 | 92.54 | 2374 | 158.31 | 3666 | 3666 |

scanning lidars since it only requires lidar specifications, which are maximum lidar range and scanner head kinematic limits (i.e., maximum speed and acceleration).

Planning the measurement campaign thoroughly especially with such complex instruments as scanning lidars ensures higher data availability during the campaign and eventually saves time and money. Lidars are very mobile and allow agile measurement
5   campaigns compared to a met mast, but too often the ease of deployment is mistaken with a limited (underestimated) need of planning. This study and the CPT, in general, show the main constraints to lidar measurements in complex terrain and give a practical solution by providing the most suitable positions where the lidar can be placed.

The point of this tool is also to carefully consider the relevance or value of using scanning lidars for a measurement campaign. In the example of the Scottish site, it is relevant to question how much improvement measuring at all turbine positions makes
10   for such a small wind farm. Is it worth using a WindScanner system instead or in addition to one met mast if we compare costs versus uncertainty in horizontal and vertical extrapolation? One way to trade off for costs is to use scanning lidars for a short



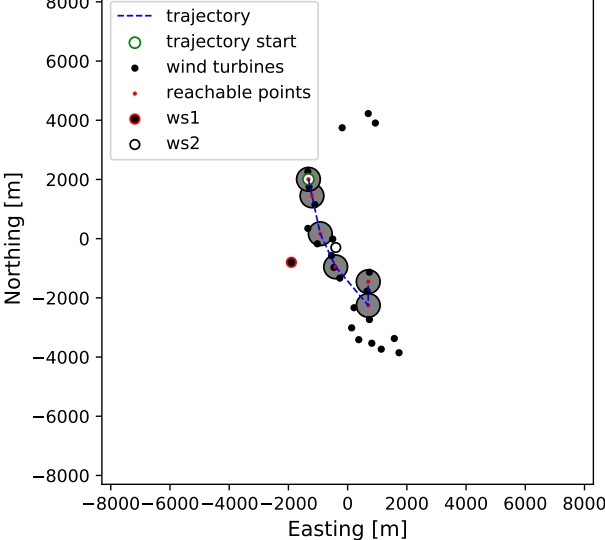

**Figure 13.** Final campaign layout for Turkish site

period, less than the 12 months required by best practices. The challenge then is the long term correction of the measurement and the related uncertainty.

This study has shown that, for a large site like the Italian or Turkish examples, one set of two WindScanners cannot measure over the whole wind farm area. This is very important to realize at the campaign planning stage when there is still time to either
give priority to one part of the site or consider using a second set of two WindScanners to cover the rest of the site.

Another major constraint that must be considered before the lidars deployment is the access roads to or near the lidar locations and possible access to a power source (e.g. existing houses, wind turbines). This is the purpose of the Satellite image used as background for the various GIS layer produced by the CPT.

In order to get around those very strong constraints, as already mentioned, it is, in any case, recommended to generate several
campaign designs and to make a site visit with thorough inspection of the possible lidar positions and verification that the data used in the CPT were accurate and up to date (e.g. obstacles, tree height).

### 4.2 Improving workflow

The presented workflow and developed tool (CPT) can already solve many important challenges regarding the scanning lidar deployment. Nevertheless, we envisage the development of several additional modules which will improve the workflow and
the developed tool.

In the current application of CPT, we were predicting the lidar range based on our experience. We plan to extend the 'Lidar range' module to be able to predict the lidar range by developing a lidar simulator. The lidar simulator will take inputs from external databases of global atmospheric visibility or aerosol optical depth for a given site and predict the expected lidar range.





Directly connected to the range prediction is the development of a module which will predict the lidar data availability at any desired range during the planned measurement period. This module will take inputs such as the predicted range from the Lidar range module as well as the cloud height, fog or mist occurrence from the WRF model to predict the data availability.

Furthermore, the module for optimizing measurement positions will be extended by considering other criteria for finding

measurement positions apart from the representativeness radius. These are for example terrain elevation, speed-up factors, roughness changes, local obstacles, etc. In principle, we will strive to incorporate anything that can cause local changes in the flow. In other words, the optimization of measurement positions will consider drivers of flow model uncertainty when finding measurement positions.

Finally, we intend to develop an eye safety module that will produce yet another restriction zone (GIS) layer for the placement

of lidars. The module will impose geometrical limitations when designing campaign layout to avoid that the laser beam is steered over the site at a height where we could expect that the human eyes can be directly exposed to it.

## 5   Conclusions

In this paper, we have provided an exhaustive description of the workflow we recommend for planning measurement campaigns using scanning lidars or WindScanner systems. The purpose is to find the most suitable positions for the lidars given

the measurement positions, the characteristics of the site (topography), the characteristics of the lidars (measurement range, kinematic limits) and the position of the two lidars relative to each other. The workflow is available through a Python library, named the Campaign Planning Tool, which will be made public during the RECAST project. The CPT has been demonstrated for planning campaigns for resource assessment on three different sites. For a small wind farm layout, the WindScanners could be placed so that measurements could be made at all turbine positions. For the other sites, that were larger, the number of

measurement points was needed to be optimized and a set of two lidars could only cover some part of the site. In any case, it is recommended to generate several possible campaign layouts and to make a site visit to take the final decision.

The CPT is easy and fast and helps to design realistic lidar measurement campaigns. Measurement campaigns are costly and risky, especially when using advanced measurement technology. The CPT helps to avoid many pitfalls that can be predicted before the start of the campaign, limiting the risks to the campaign itself.

*Author contributions.*  Conceptualization, N.V., A.V., A.B. and R.W.; Methodology, N.V. and A.V.; Software, N.V. and A.V.; Validation, N.V.; Formal Analysis, N.V.; Investigation, N.V. and A.V.; Resources, N.V. and A.B.; Data Curation, N.V.; Writing - Original Draft, N.V., A.B., R.W. and A.V.;Writing - Review & Editing, N.V., A.B., R.W. and A.V; Visualization, N.V.; Project Administration, R.W. and N.V.; Funding Acquisition, R.W. and A.B.

*Competing interests.*  Authors declare no competing interests.



*Acknowledgements.* The authors would like to acknowledge Morten Thøgersen (EMD) for his support during the conceptualization phase of the study described in the paper. The financial support for the study has been provided by the RECAST project, which is funded by Innovation Fund Denmark (7046-00021B).





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
