# Peer review of "Digitalization of scanning lidar measurement campaign planning"

_Wind Energy Science, 2019_

## Referee Comment (RC1) · Anonymous Referee #1 · 10 May 2019

**1   General comments**

In the manuscript by Vasiljevic et al. a software library is presented which allows campaign planning for wind farm site and yield assessment with Doppler wind lidar measurements. The tool seems to be a benefit for people who have experience with lidar measurements and need an initial guess for good lidar positions in the field and for these reasons the work is technically significant and important. However there are some major concerns that can be raised with regard to its scientific significance:

- Experience with meteorological measurement campaigns, especially in remote

locations, shows that logistical constraints are often dominating the site selection for instrument placement. The authors mention this issue, but only suggest to generate multiple layouts and select the one that is feasible in the end. In my opinion, the logistical constraints should be included in the selection process a priori, because it is a criterion for exclusion, while other criteria like the elevation angle and representative radius only increase the uncertainties whic could potentially be negotiated.

- To my understanding the three examples for campaign planning are not actual campaigns, but generic cases. It is not shown if the defined positions would be realistic at all, neither if the tool proofed to be efficient compared to a "normal" planning by site visits and expert knowledge. The manuscript does not show if and how the tool and process improves energy yield assessment at all.

- A great benefit would be generated if the tool allowed inexperienced users to design scanning lidar campaigns, but in multiple places in the manuscript, the authors state themselves that expert knowledge is necessary to define for example the expected range of the lidar.

- A part of the software that is very useful is the optimization of complex trajectories. I think this part is not presented very well. A mathematical desription with a definition of the variables that are included in the optimization instead of the text-based description would be much better in my opinion. I also wonder if existing python libraries (or-tools) that are available to solve traveling salesmen problems could not be applied. What is special about this problem and what makes the developed algorithm better or more suitable than others?

- In general, the manuscript is very text heavy, describing simple or trivial problems in much detail while the challenging problems are not targeted. Especially the topis mentioned in section 4.2 are scientifically challenging and significant and I think that at least one of those should be tackled in a scientific paper. A topic that

could be added to the list is the question of how many separate meausurement points are reasonable to get a represantitive average wind measurement, i.e. what is the required sampling rate?

- A major concern about the paper is that in many parts it reads more like a manual and advertisement than a scientific report and therefor could be considered inappropriate for the Wind Energy Science journal.

For all these reasons I want to encourage the authors to resubmit a manuscript that focuses on a specific research topic associated with yield assessment and lidar measurements which can be solved with that useful campaign planning tool.

**2 Specific comments**

**2.1 Introduction**

*p.2,ll.8f*: Some references should be given here. In general the introduction and manuscript are rather weak on citing relevant work.

**2.2 Section 2**

*p.3,l.12*: The *optimal* measurement positions...!?
*p.3,l.29*: Some references for the radius limits that are given should be provided.
*p.5,ll.11-19*: This seems trivial and does not need that much explanation.
*p.5-6,ll.30-10*: Public landcover maps can be quite erroneous and with a low resolution. The canopy heights can be particularly wrong, which would then lead to completely wrong results for possible lidar locations or unnecessary constraints.

*p.6,ll.29ff*: Very technical and not really relevant in this context.
*p.7,l.7*: There are many other older and peer-reviewed references for that.

**2.3  Section 3**

*Tables 1-9*: I do not think that these tables are actually necessary. The actual numbers for the measurement positions, the lidar angles etc. are irrelevant to the reader. The information that the authors want to convey should be condensed and given explicitly.
*Figures 2,6 and 10*: It is very hard to read the small white numbers in these plots. The red circle is not visible for colorblind people on green background.
*Figures 2,6 and 10*: The symbols should be a bit larger and/or in better contrast to the background.

**3  Technical corrections**

*p.1,l.1*: Strange grammar in the first sentence.
*p.1,l.2*: .. wind turbine locations.
*p.1,l.23*: I do not think that 'produce' is the right word here
*p.2,l.10*: ease of deployment
*p.2,l.15*: lays?
*p.2,l.23*: something is wrong in this sentence

---

## Referee Comment (RC2) · Anonymous Referee #2 · 21 May 2019

The manuscript "Digitizing scanning lidar measurement campaign planning" by Vasiljevic et al. introduces and describes a planning tool for finding the optimal device position for dual-Doppler lidar setups. Though I believe that this is a very relevant tool, corresponding to a major contribution by the authors, its presentation in the manuscript is not adequate for a scientific article. In many sections the text is written rather in the style of a manual than that of a paper. I strongly recommend to rearrange the manuscript, publish some of the contents in a manual-style technical report and focus in the paper on the research questions and the answers to these. Detailed comments (both technical and editorial):

Page 2, line 1 – I would like to suggest to add reanalysis date here, as a quite common option for a long-term correlation.

p. 2, l. 23 – Something wrong with the sentence "This impacts the positioning of scanning lidars can be placed..."

p. 3 l. 4 – The reference with the information in parentheses is too detailed here.

p. 3 ll. 8 – There should be no empty space in between two headings. Same for p. 9 ll. 26.

p. 3 ll. 12 – I would suggest to refer to the respective subsections within this listing.

p. 3 l. 17 – I think for a scientific paper it is not relevant that the algorithms have been developed in Python. (This really sounds as in a manual...)

p. 3 l. 17 – Here it should be briefly specified what kinds of "public databases" it is referred to.

p. 3 ll. 25 – I am missing a verb in the sentence "The approach we have used to..."

p. 4 subsection 2.3 – It is only introduced in l. 27 that a dual-Doppler setup consists of "two scanning lidars". But already in l. 21 it is referred to "one of the two lidars". Check the order of information.

p. 11 Figure 3 – I am wondering why there is so much empty space in the graphic. Is this figure really relevant, or couldn't it be combined with Figure 5.

p. 12 Figure 4 – It is rather difficult to interpret these plots, amongst others because red and white is used for two different things each.

p. 14 Table 1 and following tables – Not sure if so many details are needed for a scientific publication (I would say rather not).

---

## Author Comment (AC1) · 12 Jul 2019

**General comments**

The manuscript "Digitizing scanning lidar measurement campaign planning" by Vasiljevic et al. introduces and describes a planning tool for finding the optimal device position for dual-Doppler lidar setups. Though I believe that this is a very relevant tool, corresponding to a major contribution by the authors, its presentation in the manuscript is not adequate for a scientific article. In many sections the text is written rather in the style of a manual than that of a paper. I strongly recommend to rearrange the manuscript, publish some of the contents in a manual-style technical report and focus in the paper on the research questions and the answers to these.

[Figure]

*Dear Referee,*

*we would like to thank you for your time and for your insightful comments which were used to revise and improve our manuscript. The revised manuscript follows a classical IMRAD structure and it has now been oriented on addressing research questions instead of the description of the tool. Also, we have made a change of the title from "Digitizing scanning lidar measurement campaign planning" to "Digitalization of scanning lidar measurement campaign planning", since the term 'digitalization' better suits the work we have done .Find our detailed responses below which are provided in the italic text formating.*

**Specific comments**

Page 2, line 1 – I would like to suggest to add reanalysis date here, as a quite common option for a long-term correlation.
*Has been added*

p. 2, l. 23 - Something wrong with the sentence "This impacts the positioning of scanning lidars can be placed..."
*This sentence has been rewritten to: "This impacts the positioning of scanning lidars since we need an unobstructed passage of the beams towards measurement points. . ."*

p. 3 l. 4 – The reference with the information in parentheses is too detailed here.
*Details have been removed.*

p. 3 ll. 8 – There should be no empty space in between two headings. Same for p. 9 ll. 26.
*This is due to the LaTeX config documentclass[wes, paper]$\{copernicus\}$, which is*

*requested to be used while paper is in the review process. If we run our LaTeX file in documentclass[wes, paper]{copernicus}, i.e. configuration once the paper is approved for publication, the blank lines dissapear.*

p. 3 ll. 12 – I would suggest to refer to the respective subsections within this listing.
*The whole Section 2 in the revised manuscript has been rewritten.*

p. 3 l. 17 – I think for a scientific paper it is not relevant that the algorithms have been developed in Python. (This really sounds as in a manual. . .)
*It is important to point out that it is developed in Python, since Python is open source, and thus to use the CPT there is no investment needed (which would be the case if the tool was developed in MatLab for instance).*

p. 3 l. 17 – Here it should be briefly specified what kinds of "public databases" it is referred to.
*The whole Section 2 in the revised manuscript has been rewritten.*

p. 3 ll. 25 – I am missing a verb in the sentence "The approach we have used to. . ."
*The whole Section 2 in the revised manuscript has been rewritten.*

p. 4 subsection 2.3 – It is only introduced in l. 27 that a dual-Doppler setup consists of "two scanning lidars". But already in l. 21 it is referred to "one of the two lidars". Check the order of information.
*Dual-Doppler setups are introduced in page 2 line 17 in the reviewed manuscript.*

p. 11 Figure 3 – I am wondering why there is so much empty space in the graphic. Is this figure really relevant, or couldn't it be combined with Figure 5.

*Figure 3 has been removed in the revised manuscript*

p. 12 Figure 4 – It is rather difficult to interpret these plots, amongst others because red and white is used for two different things each.
*Red circles are used to indicate measurement point, where the addition of symbol x indicated that they are reachable by two measurement points.*

p. 14 Table 1 and following tables – Not sure if so many details are needed for a scientific publication (I would say rather not).
*In the revised manuscript we have removed tables, nevertheless data which was in tables are now provided as a supplementary material.*

---

## Author Comment (AC2) · 12 Jul 2019

**General comments**

In the manuscript by Vasiljevic et al. a software library is presented which allows campaign planning for wind farm site and yield assessment with Doppler wind lidar measurements. The tool seems to be a benefit for people who have experience with lidar measurements and need an initial guess for good lidar positions in the field and for these reasons the work is technically significant and important.

*Dear Referee,*
*We would like to thank you for your time and for your insightful comments which*

*were used to revise and improve our manuscript. We made major changes to our manuscript. The revised manuscript follows a classical IMRAD structure and it has now been oriented on addressing research questions instead of the description of the tool. Also, we have made a change of the title from "Digitizing scanning lidar measurement campaign planning" to "Digitalization of scanning lidar measurement campaign planning", since the term 'digitalization' better suits the work we have done .Find our detailed responses below which are provided in the italic text formatting.*

However there are some major concerns that can be raised with regard to its scientific significance: Experience with meteorological measurement campaigns, especially in remote locations, shows that logistical constraints are often dominating the site selection for instrument placement. The authors mention this issue, but only suggest to generate multiple layouts and select the one that is feasible in the end. In my opinion, the logistical constraints should be included in the selection process a priori, because it is a criterion for exclusion, while other criteria like the elevation angle and representative radius only increase the uncertainties which could potentially be negotiated.

*We agree with the referee that as the site constraints have large impact on the final layout of measurement campaign, especially constraints in terms of access roads and power sources. We have now explicitly stated on page 5 and 6 in the reviewed manuscript that we are creating the fifth main GIS layer (aerial image of the site) for the purpose of identifying existing road and power infrastructure. The satellite imagery is usually the first source of information a campaign planner has when he/she needs to assess whether there is necessary infrastructure for the campaign, to the very least access roads.*

*On page 11, line 7 - 11 in the reviewed manuscript we state that: "In practice, we would generate several layouts, and assess their feasibility by inspecting aerial images, e.g. looking for access roads and nearby power lines or houses."*

*To demonstrate that we followed this approach we have published the CPT outputs for the three sites in the paper.*

*Since the CPT is modular, one can use the tool in a reversed way, i.e., knowing in advance where you can place lidars and building towards identifying where you will be able to accurately perform measurements. This is now explicitly stated in Section 2.6 of the revised manuscript.*

To my understanding the three examples for campaign planning are not actual campaigns, but generic cases. It is not shown if the defined positions would be realistic at all, neither if the tool proofed to be efficient compared to a "normal" planning by site visits and expert knowledge. The manuscript does not show if and how the tool and process improves energy yield assessment at all.
*We stated in Section 3.1 of the reviewed manuscript that these are real wind farms. In the revised manuscript links to the CPT outputs for the three sites are now enclosed.*

*Using the CPT it takes roughly 5-15 minutes to generate a preliminary campaign layout, which otherwise will take longer if only Google Earth is used for this type of activity. The workflow and the tool does not exclude site visits, we have stated that in the reviewed manuscript (page 7 line 11 - 13). Actually, the workflow and tool should aid the process of site visit.*
*Also, the workflow and the tool are reducing the need for lidar expert when assessing a potential site for multi-lidar measurements.*

*Our manuscript is focused on facilitating the process of measurement campaign planning, and not about whether this process improves the AEP of future wind farm; we have adapted the abstract to avoid this confusion.*

A great benefit would be generated if the tool allowed inexperienced users to design scanning lidar campaigns, but in multiple places in the manuscript, the authors state themselves that expert knowledge is necessary to define for example the expected range of the lidar.

*The workflow and corresponding tool has been made to allow inexperienced users to design scanning lidar campaigns. Indeed, there are several input parameters which are necessary to utilize the CPT tool, specifically: representativeness radius of measurements, maximum allowed elevation angle of laser beams, minimum allowed intersecting angle between laser beams and expected lidar range. With the exception of the expected lidar range, the other three parameters have suggested values based on the existing body of knowledge. In the reviewed manuscript suggested values are stated on several locations, e.g. :*

*Representativeness radius: Page 3 Line 29*
*Maximum elevation angle: Page 2 Line 27*
*Minimum intersecting angle: Page 2 Line 29*

*We state on Page 24 Line 16 to 18 in the reviewed manuscript that we intend to extend the 'Lidar range' module to be able to perform such a task. However, this does not restrict inexperienced users of using the current version of CPT. In the revised manuscript we have provided a suggestion for the inexperienced lidar users regarding the estimation of range on Page 24 Line 17 - 21.*

A part of the software that is very useful is the optimization of complex trajectories. I think this part is not presented very well. A mathematical description with a definition of the variables that are included in the optimization instead of the text-based description would be much better in my opinion. I also wonder if existing python

libraries (or-tools) that are available to solve traveling salesmen problems could not be applied. What is special about this problem and what makes the developed algorithm better or more suitable than others?

*Following the referee's suggestion, the revised manuscript includes an improved description of the TSP.*

In general, the manuscript is very text heavy, describing simple or trivial problems in much detail while the challenging problems are not targeted. Especially the topics mentioned in section 4.2 are scientifically challenging and significant and I think that at least one of those should be tackled in a scientific paper. A topic that could be added to the list is the question of how many separate measurement points are reasonable to get a representative average wind measurement, i.e. what is the required sampling rate?

*The revised manuscript has been improved in comparison to the initial submission and also the length has been reduced. The revised manuscript follows a classical IMRAD structure and it is now oriented on addressing research questions instead of the description of the tool.*

*The topics presented in Section 4.2, which is a subsection of Discussion, outlines our future work, and thus will be the focus of our future publications.*

A major concern about the paper is that in many parts it reads more like a manual and advertisement than a scientific report and therefore could be considered inappropriate for the Wind Energy Science journal.

*See our previous comment.*

For all these reasons I want to encourage the authors to resubmit a manuscript that focuses on a specific research topic associated with yield assessment and lidar measurements which can be solved with that useful campaign planning tool.

*We have taken into account the referee's suggestions and improved our manuscript.*

**Specific comments**

2.1 Introduction
p.2,ll.8f: Some references should be given here. In general the introduction and manuscript are rather weak on citing relevant work.
*The introduction contains citation to 14 communications related to the topic that the paper addresses. Nevertheless, we are eager to improve the introduction and therefore, we kindly ask the referee to suggest a list of references that needs to be reviewed and cited in the introduction.*

2.2 Section 2
p.3,l.12: The optimal measurement positions...!?
*The whole Section 2 in the revised manuscript has been rewritten.*

p.3,l.29: Some references for the radius limits that are given should be provided.
*We have added a references (MEASNET) to the line stating the radius limits.*

p.5,ll.11-19: This seems trivial and does not need that much explanation.
*The whole Section 2 in the revised manuscript has been rewritten.*

p.5-6,ll.30-10: Public landcover maps can be quite erroneous and with a low resolution. The canopy heights can be particularly wrong, which would then lead to completely wrong results for possible lidar locations or unnecessary constraints.
*We agree, they are however often the starting point for any resource assessment*

*before site visits are conducted. Conservative land-cover translations, e.g. using tall tree-heights are recommended initially and the data can subsequently be corrected by the site engineers after consulting aerial imagery or conducting a site visit.*

p.6,ll.29ff: Very technical and not really relevant in this context.
*The whole Section 2 in the revised manuscript has been rewritten.*

p.7,l.7: There are many other older and peer-reviewed references for that.
*We have added the oldest reference we found when comes to the dual-radar/dual-Doppler measurements setup in the list that is Davies-Jones 1979.*

2.3 Section 3
Tables 1-9: I do not think that these tables are actually necessary. The actual numbers for the measurement positions, the lidar angles etc. are irrelevant to the reader. The information that the authors want to convey should be condensed and given explicitly.Figures 2,6 and 10: It is very hard to read the small white numbers in these plots. The red circle is not visible for colorblind people on green background.
Figures 2,6 and 10: The symbols should be a bit larger and/or in better contrast to the background. *In the revised manuscript we have removed tables, nevertheless data which was in tables are now provided as a supplementary material. The commented figures have been improved.*

**Technical corrections**

p.1,l.1: Strange grammar in the first sentence.
*The abstract has been modified.*

p.1,l.2: .. wind turbine locations.

*Corrected accordingly.*

p.1,l.23: I do not think that 'produce' is the right word here
*The sentence has been rewritten to:*
*The local measurements are used to produce the observed wind climate of the site.*

p.2,l.10: ease of deployment
*Corrected accordingly.*

p.2,l.15: lays?
*'lays' replaced with 'lies'*

p.2,l.23: something is wrong in this sentence
*The sentence has been rewritten to:*
*This impacts the positioning of scanning lidars since we need an unobstructed passage of the beams towards measurement points, i.e. clear line-of-sights (LOS)*
* * *

---

## Referee Report (RR1)

Dear Nikola, Andrea, Andreas and Rozenn,

Your manuscript "Digitalization of scanning lidar measurement campaign planning" tackles an important topic and the paper and the corresponding open source tool are a very valuable contribution to the lidar community. Scanning lidar campaign planning is tricky and (as you also point out) requires expert knowledge. I see your CPT as a contribution to a lidar community toolbox that digitalizes knowledge from a few lidar expert heads and thus will lower the hurdle for inexperienced end users to apply scanning lidars for resource assessment. Plus it enables experts to work together on a common platform to further improve how we use our lidars.

Reading the comments of the reviewers of your first manuscript version, the main concern was that it reads more like a manual to the tool and does not explain the scientific content well enough. In the meantime you have revised the paper thoroughly and explained the scientific contribution in more detail. In my opinion your manuscript is now almost ready for publication. Below I suggest some minor revision and technical corrections.

Ines

Specific comments

- p.1 l.16 f: Your last sentence of the abstract is a bit weak. Why is it only important whether the site can be covered by one system or not? I suggest adding a sentence that goes more like "With minimal effort, the CPT is able to optimize lidar measurement positions and suggest possible lidar installation positions for carrying out a resource assessment campaign. Thus it shows for instance instantly whether the whole site can be covered by one system or not".
- p.1 l.19 f Introduction: Nikola, it would be nice to have a reference here to your paper "Perdigão 2015: methodology for atmospheric multi-Doppler lidar experiments". Which steps of your campaign planning methodology are you covering with the CPT?
- p.3 l.21 f: Please include the sentence "We assume that the wind farm site has been selected and that a preliminary resource assessment and wind farm layout have been made prior to the campaign planning." again. This was necessary information.
- You could structure the sections describing the phases of the CPT better: 1) What are the challenges that occur when planning a lidar campaign in that phase? 2) What are the solutions that you found and that the CPT is offering? 3) How is it implemented in the tool? Most sections you start with 2) or 3).
  p.3.l.21ff: E.g. for phase 1 you start the section by saying "The wind farm layout is a required input for the campaign planning workflow". Instead you could start with "When planning a lidar measurement campaign, the first challenge is to determine where the lidar should measure. For a wind resource assessment campaign for a future wind farm, the goal is to measure wind speed and –direction at hub height of the turbines. In the CPT we assume that the wind farm site has been selected and that a preliminary resource assessment and wind farm layout have been made prior to the campaign planning. Thus the wind farm layout is a required input for the campaign planning workflow." Then continue explaining (as you did) that in the best case you measure at every turbine, but the number of measurement points is restricted and you need to find a solution when you have too many turbines and so on..
- p.5 l.8 f: add the description of the variable M: "[…] calculate a midpoint *M*, […]"
- p.6 l. 29: what does CLC stand for? CORINE Land Cover? Then introduce the abbreviation in l.17.

- p.6 l. 39: Is there a reference for the CLC code or did you come up with those numbers yourself? I was wondering if an overview table for the different land cover types was helpful, but this might be overkill and go back to the manual style. But at least a reference where the reader can look up the code would be helpful.
- P.8 Figure 3: To read it, I printed out your paper in black/white and then couldn't differentiate between red/green. Something to keep in mind when choosing colours, as there are also people with red-green blindness.
- p. 8 l 10f: A note here would help that the expected range of the lidar is not the maximum range given in the product data sheet. I know you explain it later when you apply the tool, but nevertheless, it's worthwhile mentioning here as well.
- General question to phase 2&3: Shouldn't the placement of the lidars also depend on the main wind direction, as we know that LOS measurement directions perpendicular to the wind should be avoided? This might be an additional layer and would require prior information of the wind conditions at the site. Could be something for a future update of the tool.
- p.10 l.20: Explain what theta and phi are
- p.10 l.21: The set T is not related to the turbine position set T from phase 1 is it? Please explain your variables in this section better as this leads to confusion.
- p.11 l.20: Explain what theta and phi are
- p.12 Figure 4: explain that *ws* stands for wind scanner
- p.13 l.2f: You mention that "current commercial scanning lidars allow only step-stare implementation of complex trajectories". This is actually not true – our StreamLine scanner can do continuous measurements for arbitrary trajectories. However, the synchronisation of two devices in the continuous mode is very difficult and step-stare makes more sense anyway if you want to measure at a specific point.
- P.13 l.5f: Note: for our StreamLine lidar scanner, the maximum acceleration of the elevation motor is actually higher than for the azimuth motor. This means you should calculate T_move for the elevation and the azimuth motion and then take the higher value to calculate the required time for the movement. You could note in the text on that.
- p.13 l.20: I'm missing an explanation after that first sentence, why exactly those three sites are of interest to test the CPT. How do they differ from each other?
- p.14 Figure 5: do you need a source reference here to Google? Same for Figure 8 and 12
- p.14 l.5: How long does it take to generate one measurement campaign layout? Seconds? Minutes? In general I think it would be very interesting to know how long it takes to run the CPT. This would emphasise that it is a very useful tool to get a first quick idea on how to set up the campaign.

Technical corrections

- p.1 l.2 f: […] the extrapolation distance […] *is* reduced
- p.2 l.29: […] with the vertical component*. F*inally, the intersecting beams […]
- p.2 l.34: […] resulting digital tool named Campaign Planning Tool *(*CPT)**,** […]
- p.3 l.4:  *In addition*, […]
- p.11 l.19: […] can *be* seen […]
- p.14 l.5 the two commas around the "therefore" are overkill
- p.15 Figure 6 label: […] first lida*r* at […]
- p.16 l. 15: […] farmland, *al*though in […] ("though" is colloquial in my opinion)
- p.21 l. 3: *As* for the […]
- p.24 l. 4: […] large site *such as* the […]
- p.24 l.32: […] designing *the* campaign layout […]

---

## Author Response (AR2)

**Replies to referees**

Nikola Vasiljević

**1 Referee 1**

**General comments**

In the manuscript by Vasiljevic et al. a software library is presented which allows campaign planning for wind farm site and yield assessment with Doppler wind lidar measurements. The authors have revised the manuscript and put more emphasize on describing the algorithms that were developed to optimize lidar location and scanning trajectories. This makes the manuscript more suitable for a scientific journal such as Wind Energy Science. There are some points that should still be improved before publication in my opinion:

- The authors have changed the structure and content of Sections 2 and 3 in what I think is a good way. They did however not transfer these changes into the introduction, discussion and conclusion sections. Only very few changes have been made in these sections.

- In the discussion and conclusion I would expect some kind of an evaluation of how much the algorithms can improve availability of measurement points compared to a non-optimized set-up, or mast measurements. Some prediction or estimation of the improvement in wind resource assessment of the case studies would be even more valuable.

- For the trajectory planning, some variants of trajectories are shown in Section 2, but it is not evaluated how large the gain in measurement speed is. This could for example be done for the example sites in Section 3 and discussed in Section 4. Possibly, this would also allow some conclusions under which conditions the CPT is most valuable.

- The conclusion section still has the character of an advertisement for a software tool, which is not appropriate for a scientific journal.

*Dear Referee,*

*We would like to thank you for your time and for your insightful comments which were used to revise and improve our manuscript. We made several important changes to our manuscript which addresses issues raised in your second review of our paper. We would like to point out that our paper is now primarily focused on the workflow, methodology and algorithms and not on the software. Nevertheless, in mean time between the first and second revision we have publicly released the Python package which represents the digital version of the presented workflow. Furthermore, all results presented in the reviewed paper are fully reproducible since scripts that were used to produce them are now publicly available as a supplementary material. Find our detailed responses below which are provided in the italic text formatting.*

The authors have changed the structure and content of Sections 2 and 3 in what I think is a good way. They did however not transfer these changes into the introduction, discussion and conclusion sections. Only very few changes have been made in these sections.

*We have revised the three sections that the referee refers to. Section 1 now provides a better introduction to the challenges that*
5  *the paper is addressing. Similarly, Section 4 and 5 have an improved discussion and conclusion of the solutions that the paper provides.*

In the discussion and conclusion I would expect some kind of an evaluation of how much the algorithms can improve avail-
10  ability of measurement points compared to a non-optimized set-up, or mast measurements. Some prediction or estimation of the improvement in wind resource assessment of the case studies would be even more valuable.

*Section 3 now contains estimation of the AEP uncertainty using Clerc et al.(2012) model for a centrally located mast and multi-point WindScanner measurements. The discussion of these results are presented in Section 4.*

15  For the trajectory planning, some variants of trajectories are shown in Section 2, but it is not evaluated how large the gain in measurement speed is. This could for example be done for the example sites in Section 3 and discussed in Section 4. Possibly, this would also allow some conclusions under which conditions the CPT is most valuable.

*In the revised manuscript Section 3 provides calculation of how much the trajectory timing is improved using the presented workflow and software tool in comparison to manual approach. Then again, these results are discussed in Section 4 and con-*
20  *cluded in Section 5. In our opinion one should always run the optimization of trajectory since this process does not take much time and it guarantees an improved measurement rate.*

The conclusion section still has the character of an advertisement for a software tool, which is not appropriate for a scientific journal.

25  *The revised manuscript contains an improved conclusion which is more in accordance to the style of classical scientific journals. There is no mentioning of the software package in the conclusion, except indications about the digitilized version of the workflow.*

**2 Referee 3: Ines Wuerth**

**General comments**

30  Dear Nikola, Andrea, Andreas and Rozenn, Your manuscript "Digitalization of scanning lidar measurement campaign planning" tackles an important topic and the paper and the corresponding open source tool are a very valuable contribution to the lidar community. Scanning lidar campaign planning is tricky and (as you also point out) requires expert knowledge. I see your CPT as a contribution to a lidar community toolbox that digitalizes knowledge from a few lidar expert heads and thus will

lower the hurdle for inexperienced end users to apply scanning lidars for resource assessment. Plus it enables experts to work together on a common platform to further improve how we use our lidars. Reading the comments of the reviewers of your first manuscript version, the main concern was that it reads more like a manual to the tool and does not explain the scientific content well enough. In the meantime you have revised the paper thoroughly and explained the scientific contribution in more detail. In my opinion your manuscript is now almost ready for publication. Below I suggest some minor revision and technical corrections. Ines

*Dear Ines,*

*We would like to thank you for your time and for your insightful comments which were used to revise and improve our manuscript.*

**Specific comments**

p.1 l.16 f: Your last sentence of the abstract is a bit weak. Why is it only important whether the site can be covered by one system or not? I suggest adding a sentence that goes more like "With minimal effort, the CPT is able to optimize lidar measurement positions and suggest possible lidar installation positions for carrying out a resource assessment campaign. Thus it shows for instance instantly whether the whole site can be covered by one system or not".

*The last sentence has been rewritten according to the referee suggestion.*

p.1 l.19 f Introduction: Nikola, it would be nice to have a reference here to your paper "Perdigao 2015: methodology for atmospheric multi-Doppler lidar experiments". Which steps of your campaign planning methodology are you covering with the CPT?

*The introduction now clearly states that the workflow addresses 'experiment layout design' and 'scanning modes design' steps of the Perdigao-2015 methodology.*

p.3 l.21 f: Please include the sentence "We assume that the wind farm site has been selected and that a preliminary resource assessment and wind farm layout have been made prior to the campaign planning." again. This was necessary information.

*The manuscript has been updated accordingly.*

You could structure the sections describing the phases of the CPT better: 1) What are the challenges that occur when planning a lidar campaign in that phase? 2) What are the solutions that you found and that the CPT is offering? 3) How is it implemented in the tool? Most sections you start with 2) or 3). p.3.l.21ff: E.g. for phase 1 you start the section by saying "The wind farm layout is a required input for the campaign planning workflow". Instead you could start with "When planning a lidar measurement campaign, the first challenge is to determine where the lidar should measure. For a wind resource assessment campaign for a future wind farm, the goal is to measure wind speed and –direction at hub height of the turbines. In the CPT we assume that the wind farm site has been selected and that a preliminary resource assessment and wind farm layout have been made prior

to the campaign planning. Thus the wind farm layout is a required input for the campaign planning workflow." Then continue explaining (as you did) that in the best case you measure at every turbine, but the number of measurement points is restricted and you need to find a solution when you have too many turbines and so on.

*We followed the suggestion by the reviewer and made the modification of the manuscript such that each phase of the workflow*
5 *clearly states what challenges it is addressing.*

p.5 l.8 f: add the description of the variable M: "[...] calculate a midpoint M, [...]"
*The manuscript has been revised accordingly.*

10 p.6 l. 29: what does CLC stand for? CORINE Land Cover? Then introduce the abbreviation in l.17.
*It is actually grid_code and not CLC that we are using. Accordingly we made the modification in our manuscript.*

p.6 l. 39: Is there a reference for the CLC code or did you come up with those numbers yourself? I was wondering if an overview table for the different land cover types was helpful, but this might be overkill and go back to the manual style. But at
15 least a reference where the reader can look up the code would be helpful.
*The reference to the web page of the Corine Land Cover dataset is now provided in the manuscript. The look-up table is a part of the CLC dataset (in zip file as an Excel table).*

P.8 Figure 3: To read it, I printed out your paper in black/white and then couldn't differentiate between red/green. Some-
20 thing to keep in mind when choosing colours, as there are also people with red-green blindness.
*The figure is changed, and instead of red and green we use grey and white color to highlight cells.*

p. 8 l 10f: A note here would help that the expected range of the lidar is not the maximum range given in the product data sheet. I know you explain it later when you apply the tool, but nevertheless, it's worthwhile mentioning here as well.
25 *The manuscript was revised according to the referee suggestion.*

General question to phase 23: Shouldn't the placement of the lidars also depend on the main wind direction, as we know that LOS measurement directions perpendicular to the wind should be avoided? This might be an additional layer and would require prior information of the wind conditions at the site. Could be something for a future update of the tool.
30 *The current workflow is a wind direction independent due to the demand on the intersecting angle between two laser beams. However, indeed the workflow could be optimized to be direction dependent which would in principle reduce the demand on the intersecting angle.*

p.10 l.20: Explain what theta and phi are
35 *The explanation of the symbols is provided in the revised manuscript.*

p.10 l.21: The set T is not related to the turbine position set T from phase 1 is it? Please explain your variables in this section better as this leads to confusion.

*T is a set of measurement points, which often coincides with turbine hub position.*

p.11 l.20: Explain what theta and phi are

*The explanation of the symbols is provided in the revised manuscript.*

p.12 Figure 4: explain that ws stands for wind scanner

*Figure 4 caption states that ws1 and ws2 are names for the two lidars.*

p.13 l.2f: You mention that "current commercial scanning lidars allow only step-stare implementation of complex trajectories". This is actually not true – our StreamLine scanner can do continuous measurements for arbitrary trajectories. However, the synchronisation of two devices in the continuous mode is very difficult and step-stare makes more sense anyway if you want to measure at a specific point.

*The indicated sentence has been removed and suggestions of the referee incorporated in the revised manuscript.*

P.13 l.5f: Note: for our StreamLine lidar scanner, the maximum acceleration of the elevation motor is actually higher than for the azimuth motor. This means you should calculate $T_{move}$ for the elevation and the azimuth motion and then take the higher value to calculate the required time for the movement. You could note in the text on that.

*The revised manuscript now provides suggestions what to do in case when kinematic limits are different among rotational axes of the scanner head.*

p.14 Figure 5: do you need a source reference here to Google? Same for Figure 8 and 12

*Indeed. The Figure captions now indicate the source of the aerial images.*

p.14 l.5: How long does it take to generate one measurement campaign layout? Seconds? Minutes? In general I think it would be very interesting to know how long it takes to run the CPT. This would emphasise that it is a very useful tool to get a first quick idea on how to set up the campaign.

*It takes about a few minutes to design the campaign mainly due to some manual work an end-user needs to do (choosing the first and second lidar positions). Otherwise, on an average computer it takes between 10 to 40 seconds to run all the processes. The length of the execution is dependent on the wind farm size and number of wind turbines.*

[revised manuscript text omitted]